# Assessing the Biofilm Formation Capacity of the Wine Spoilage Yeast *Brettanomyces bruxellensis* through FTIR Spectroscopy

**DOI:** 10.3390/microorganisms9030587

**Published:** 2021-03-12

**Authors:** Maria Dimopoulou, Vasiliki Kefalloniti, Panagiotis Tsakanikas, Seraphim Papanikolaou, George-John E. Nychas

**Affiliations:** Laboratory of Microbiology and Biotechnology of Foods, Department of Food Science and Human Nutrition, School of Food and Nutritional Sciences, Agricultural University of Athens, Iera Odos 75, 11855 Athens, Greece; maroula.dimo@hotmail.com (M.D.); kefalloniti.v@gmail.com (V.K.); p.tsakanikas@aua.gr (P.T.); spapanik@aua.gr (S.P.)

**Keywords:** *Brettanomyces bruxellensis*, biofilm formation, FTIR analysis, machine learning, classification, wine spoilage

## Abstract

*Brettanomyces bruxellensis* is a wine spoilage yeast known to colonize and persist in production cellars. However, knowledge on the biofilm formation capacity of *B. bruxellensis* remains limited. The present study investigated the biofilm formation of 11 *B. bruxellensis* strains on stainless steel coupons after 3 h of incubation in an aqueous solution. FTIR analysis was performed for both planktonic and attached cells, while comparison of the obtained spectra revealed chemical groups implicated in the biofilm formation process. The increased region corresponding to polysaccharides and lipids clearly discriminated the obtained spectra, while the absorption peaks at the specific wavenumbers possibly reveal the presence of β-glucans, mannas and ergosterol. Unsupervised clustering and supervised classification were employed to identify the important wavenumbers of the whole spectra. The fact that all the metabolic fingerprints of the attached versus the planktonic cells were similar within the same cell phenotype class and different between the two phenotypes, implies a clear separation of the cell phenotype; supported by the results of the developed classification model. This study represents the first to succeed at applying a non-invasive technique to reveal the metabolic fingerprint implicated in the biofilm formation capacity of *B. bruxellensis,* underlying the homogenous mechanism within the yeast species.

## 1. Introduction

*Brettanomyces bruxellensis* is non-conventional wine yeast, with remarkable spoilage potential. The spoilage effect is due to the high level of production of undesirable volatile phenols through the metabolism of the hydroxycinnamic acids in grapes, which have a detrimental and irreversible effect on the wine’s aroma [1,2,3,4,5]. Additionally, the capacity of the species to enter under oenological conditions into a viable but not cultivable (VBNC) state shows that the frequently used culture dependent methods could create a false estimation of the yeast population, complicating the detection of yeast in production facilities [6,7]. The fact that *B. bruxellensis* exhibits an elevated ability to invade wine cellars and contaminate winery equipment, represents an area of increasing concern for the wine industry [8,9]. The occurrence of *B. bruxellensis* in winemaking equipment highlights the colonization and the biofilm formation capacity of the species in the winemaking environment, and consequently, demonstrates the persistence of the yeast in the cellar from one year to the next [8,10].

Biofilms are highly structured multicellular aggregates that are able to adhere to and grow on biotic and abiotic surfaces [11,12,13]. This complex structure is mainly composed of cell-produced extracellular polymeric substances (EPS), which are essential for biofilm formation. EPS comprise a wide variety of polysaccharides, proteins, glycoproteins, as well as glycolipids and extracellular DNA (e-DNA) [14]. So far, the methods used for the identification of the EPS are quite invasive and the extraction protocols usually target specific molecules of interest and not the whole matrix [12].

Fourier transform infrared (FTIR) spectroscopy is a powerful, fast and non-invasive technique for generating direct information about the molecular and chemical composition of the analyzed sample, based on the absorption of infrared radiation [15]. The technique is based on the vibration of atoms when the IR radiation passes through the sample, which generates a vibrational spectra specific to each compound [13]. The method is widely applied to discriminate biological samples due to the fact that different cell components produce different absorption profiles [16]. More precisely, FTIR has been used for the classification and identification of microorganisms at the species and strain levels [17,18]. Interestingly, the method has proven to be capable of identifying species of clinical samples for diagnostic reasons or for epidemiological surveillance [19]. In the food industry, the metabolic fingerprint obtained from the FTIR spectra can be used for rapid food identification, detection and prediction of microbial spoilage [20,21,22,23]. Additionally, the FTIR spectra analysis can record the biochemical compounds excreted by the microorganisms, such as those implicated in the EPS biofilm matrix. For instance, FTIR was recently successfully applied to characterize the chemical composition of the biofilm matrix implicated in dental plaque formation by streptococci species [24]. Respectively, by the same reasoning, FTIR can also be applied to predict the microbial mode of life in planktonic or embedded growth.

The aim of the present study was to deepen our knowledge on the biofilm formation capacity of *B. bruxellensis* on stainless steel surfaces by applying FTIR spectroscopy. For this reason, the main parts of the study were to (i) investigate the adhesion capacity of eleven strains of the species, (ii) compare the metabolic fingerprint of the attached vs. planktonic cells, (iii) reveal and discuss the compounds of interest implicated in the biofilm formation capacity of the species associated with wine spoilage.

## 2. Materials and Methods

### 2.1. Yeast Strains and Growth Condition

*B. bruxellensis* strains previously isolated from oenological materials [25] were kindly provided from the ARS Culture Collection (NRRL) and Institute of Oenology of Bordeaux (IOEB) collection, as well as from the collection of the Department of Food Science and Technology of Aristotle University of Thessaloniki. The 11 strains belonged to six different genetic groups based on the analysis of 12 microsatellites markers especially developed for the species [26].

### 2.2. Biofilm Formation on Stainless Steel Coupons

The 11 strains of *B. bruxellensis* were tested for their biofilm formation capacity on stainless steel coupons. Microorganisms were stored at −80 °C in vials with glycerol (30%) and were activated by adding 200 μL to 10 mL Yeast Peptone Dextrose (YPD) medium at 28 °C for 48 h. Then, for the working cultures, a 100 μL suspension of the activated cells was inoculated in 10mL of YPD and left for 72 h until reaching the late exponential phase. 

Cells for the working cultures were rinsed twice (4000× *g*, 4 °C, 10 min) with sterile Ringer solution (tablets Merck KGaA, Darmstadt, Germany) and re-suspended in Ringer solution in order to obtain an OD_600 nm_ = 1. A total of 0.5 mL of the cell suspension was added to a test tube containing 4.5 mL of Ringer solution and one coupon of stainless steel (3 × 0.8 × 0.1 cm type AISi-304), reaching an inoculation rate of 10^7^ cfu/mL. A 5 mL amount of the final cell suspension in Ringer solution completely covered the coupon surface and was left at 25 °C for 3 h. After the incubation period, 1 mL of the culture was used to estimate the population of planktonic cells (10-fold serial dilutions on YPD agar plates) while the coupon was carefully removed with sterile forceps to a new tube and washed on both sides with Ringer solution in order to remove all the loosely attached cells. After the washing step, the coupon was transferred to a new falcon tube containing 6 mL of Ringer solution and 10 sterile glass beads (3 mm) covered with silicon to perform the bead vortex method. Each falcon tube containing the coupon was vigorously swirled (vortex) for 2 min. After vortexing, 1 mL of suspension was used to estimate the attached cultivable cell population (10-fold serial dilutions on YPD agar plates). Each experiment was performed in triplicate and repeated independently two times.

### 2.3. Fourier Transform Infrared Spectroscopy

The infrared spectra of the cell suspension samples were recorded by a ZnSe 45o HATR (Horizontal Attenuated Total Reflectance) crystal (PIKE Technologies, Madison, WI, USA) and by a FTIR-6200 JASCO spectrometer (Jasco Corp., Tokyo, Japan). Spectra ManagerTM Code of Federal Regulations software v. 2 (Jasco Corp.) was used to collect the obtained spectra (wavenumber range of 4000 to 400 cm^−1^).

After 3 h of incubation in contact with the coupon surface, 1 mL of culture of the 11 *B. bruxellensis* strains was retrieved and transferred to the crystal plate in order to obtain the FTIR spectra of the planktonic cells. Then, after performing the bead vortex method for dispatching the attached cells from the formed biofilm on the coupon surface, 1 mL of suspension was transferred to the crystal plate to obtain the attached cells spectra, respectively. The process is illustrated in Figure 1. Prior to the first sample measurement, the reference spectra of the crystal were obtained and this process was repeated after every five sample measurements. Between measurements, the crystal was cleaned with distilled water and acetone, while it was left to dry completely until the next sample addition.

### 2.4. FTIR Spectra Data Analysis

Herein, a detailed description is provided of the analysis pipeline and the corresponding justification (please refer to Figure 2). The data processing and analysis pipeline consists of data pre-processing and normalization, followed by the extra-tree method [27] (standing for extremely randomized trees), employed specifically for supervised dimensionality reduction on the basis of informative, important wavenumber selection. After the wave numbers were selected and the problem was limited to a small space (fewer features), we followed two different approaches in order to essentially reach the same target. Concerning the first approach, we employed unsupervised clustering of the data using principal component analysis (PCA) [28] and Gaussian mixture modeling [29] in order to identify if the two clusters of planktonic and adhered samples appeared as the “natural” clusters of the data. At the second approach we follow a classification strategy where the selected wavenumbers have been used as inputs to an SVM [30] classifier in order to be trained and used as a model for sample prediction, decision making. 

As shown in Figure 2, the first data manipulation process is to truncate the noisy areas at the start of the spectra and the area above ~3100 cm^−1^, corresponding to the absorption of water (any information under this peak is hidden and cannot be extracted) prior to normalization (refer to Appendix A). Thus, we have the range ~880–3100 cm^−1^. Next in the workflow, and in order to enhance the quality of the data while also reducing the correlated information across the different wavenumbers and eliminate the inherent multiplicative noise, the robust normal variate [31] (RNV) model—which the the robust version of the standard normal variate [32] (SNV) model—is employed so as to ensure improved downstream analysis (please refer to Appendix A). Specifically, the RNV is given by: (1)Sisnv=si−median(S)/ mad(S)
where *S* is the ensemble of all spectra, and *si* and *sisnv* are the ith and the corresponding normalized spectra, respectively. *median* absolute deviation [33] (*mad*) is a robust measure of the variability of a univariate sample of quantitative data s1,s2,…,sn computed as: (2)mad=median(si)−median(S)

As the next step, dimensionality reduction via feature selection was performed. For this, our method of choice was the extra-trees method, which we used in order to identify the most important and informative wavenumbers for classifying the planktonic vs. biofilm state/phenotype. This step is critical since in this study we have 116 samples and each sample—after the aforementioned pre-processing steps—counts for 2300 features. Thus, by limiting the number of variables, we were able to avoid overfitting, downgrading to a lower dimension space, while simplifying our problem both for clustering and classification. First, we split the data (129 samples, where 63 were from biofilm and 66 were from planktonic forms) randomly into training (99) and test (30; 16 planktonic and 14 biofilm) groups. The extra-trees was trained with 50 trees in the forest, five-fold cross validation and the function to measure the quality of a split was “gini” for the Gini impurity. Gini impurity is a measure of how often a randomly chosen element from the set would be incorrectly labelled if it was randomly labelled according to the distribution of labels in the subset. Random forest analysis has been previously used in food analysis [23,34,35,36]. Herein, we turn to extra-trees, where the main objective is to further randomize tree building in the context of numerical input features, where the choice of the optimal cut-point is responsible for a large proportion of the variance of the induced tree. In comparison to random forests, the extra-trees method drops the idea of using bootstrap copies of the learning sample and instead of trying to find an optimal cut-point for each one of the K randomly chosen features at each node, it selects a cut-point at random. In the productive context of having many problems characterized by a large number of numerical features varying more or less continuously, as we have here, the extra-trees method can lead to increased accuracy due to its smoothing, while it simultaneously has a lower computation cost as it does not rely on the determination of optimal cut-points as is the case in standard trees and in random forests. Furthermore, leaving the bootstrapping idea can lead to advantages in terms of bias, whereas the cut-point randomization method often results in an excellent variance reduction effect [27]. Finally, from a functional point of view, the extra-trees method produces piece-wise multilinear approximations, rather than the piece-wise constant ones of the random forest method. Therefore, for all of these reasons, we believe that the adopted approach presented herein will provide us with robust features for efficient classification and clustering purposes. 

Under the scope of the explanatory data analysis via unsupervised clustering, we used the data (129 FTIR spectra samples) with a reduced number of features, as discussed previously. As the next step and in order to visualize the data, we further reduced the dimensionality by principal component analysis (PCA). The first two principal components accounted for 99.4% of the total variance of the data and we kept only these first two components. On the two-dimensional space of PC1 and PC2, we applied Gaussian mixture modeling with two clusters, as this is the number of known states of *B. bruxellensis*. 

Concerning the classification analysis pipeline, SVM-based modeling was employed with the use of the reduced dimensionality data and the output of the extra-trees method. Support vector machines (SVMs) are models that are created in a supervised manner and are used for classification and regression analyses [37]. SVMs are a well suited classification technique when the training data consist of large number of variables in relation to the number of observations. In our case, the input matrix X contains 99 normalized FTIR training samples with 45 variables/wavenumbers each, while the Y matrix is a single column matrix consisting of the class (coded as a number, 0 and 1) for each corresponding sample state, i.e., attached or planktonic. A five-fold cross-validation method was used, with shuffling of the training set, and it was then split into five subsets (folds). A grid search [38] was also applied for determining the optimum parameters for the SVM model. A grid search is an exhaustive search through a manually specified subset of the parameter space, combined with cross validation, in an attempt to compute the ideal kernel and parameters for the SVM classification. Herein, the kernels tested were: linear, radial basis function (rbf) and polynomial, the search range for C parameter was set as (0.0001, 0.001, 0.01, 0.1, 1, 10, 100, 1000), while the γ parameter (10–6, 10–1) was in a logarithmic scale and the degree = 2, 3, 4 and 5. The outcomes for the hyper-parameters grid search resulted in choosing the linear kernel and C = 10, as the optimal classifier parameters for our data. Afterwards, the SVM was trained using the resulting optimal kernel and corresponding parameters. 

The whole workflow was implemented in python 3.7, employing the scikit-learn library [38,39]. All codes and data used herein are available upon request from the authors.

### 2.5. Statistical Analysis

Results are expressed as mean ± standard deviation of the independent experiments and analysis of variance (ANOVA), followed by post hoc Tukey’s test (HSD), was used for comparisons. A *p*-value ≤ 0.05 was considered statistically significant.

## 3. Results

### 3.1. Biofilm Formation Capacity of B. bruxellensis Strains on Stainless Steel Coupons

Strain variability has often been observed for the studied species under the various phenotypic traits and for the particular case of biofilm formation [7,8,37]. In this study, 11 strains from different genetic groups were chosen based on previous studies that tested the population structure of approximately 1510 isolates of *B. bruxellensis* from 29 countries. Based on microsatellite analysis, the tested isolates were grouped into six main genetic groups, named by the reference strain of each group: strains AWRI1608 and GSP1516 (AWRI 1608-like), strains CBS 2499 and CBS 78 (2499-like), strains 33.1 and 33.3 (AWRI 1499-like), CBS5512 and CBS 6055 (CBS5512-like), strain UWOPS92.244.4 (L0308-like and L14165-like), and strains 15.12 and 45.3 (L 14165-like) [40,41]. 

The biofilm formation capacity of the 11 *B. bruxellensis* strains on a stainless steel surface in an aqueous solution was monitored in six independent biological replicates, with an incubation period of 3 h. All the strains were inoculated at the same population level of 10^7^ cfu/mL. Statistical comparisons separated the strains into three groups based on their adhesion capacity (Figure 3). The strain 33.3 presented the smallest attached cell population on the coupons (log_10_ 2.32 ± 0.29 cfu/cm^2^), reaching statistical significance. Seven strains belonged to the group that expressed an intermediate adhesion capacity, with an average population of attached cells of log_10_ 3.18 ± 0.54 cfu/cm^2^. Three strains differed from the rest due to their high concentration of attached cells on the coupon surface: 33.1 (log_10_ 5.6 ± 0.29 cfu/cm^2^), CSB6055 (log_10_ 5.43 ± 0.24 cfu/cm^2^) and 15.12 (log_10_ 5.25 ± 0.15 cfu/cm^2^).

### 3.2. FTIR Spectra Analysis of Attached vs. Planktonic Cells of B. bruxellensis 

In this subsection, the results and findings of the two-way analysis of the FTIR spectra are presented, considering two states of the cells of *B. bruxellensis* for all 11 strains used herein. Following the first type of analysis, i.e., via unsupervised clustering (as discussed in the Material and Methods Section), we conclude that, given a set of measurements, we are able to identify two distinct populations forming two “natural” inherent classes that correspond to the distinction of attached cells vs. the planktonic cells. This result—also depicted in Figure 4 with the two mode mixture of Gaussians—is a very interesting outcome, showing that attached cells exhibit much less deviation among them even when the input spectra originate from different strains and cultures (11 strains and six independent cultures in our case). On the other hand, planktonic cells exhibit a greater level of variability in the PC space, but the two clusters are still easily separable along the PC1. This outcome could be used in the development of a model to cluster any new sample to one of these clusters, thus providing an efficient and reliable, non-invasive and direct decision-making approach, regardless of the specific strain and culture. The difference of the two cell states in the PC space can be explained if we consider the major difference between the two cell states, i.e., attached vs. planktonic cells. In the case of biofilm colony formation, this colony comprises a consortium of cells that stick to each other and to a surface [42]. These attached cells become embedded within a slimy extracellular matrix that is composed of extracellular polymeric substances (EPS), which are a polymeric conglomeration of extracellular polysaccharides, proteins, lipids and DNA [43,44]. Thus, these EPSs are expected to be less variant due to the differences among the strains and so this can explain the low variance of the Gaussian mode, representing the adhered/biofilm samples, as presented in Figure 4. 

### 3.3. Chemical Compound Groups Discriminate Planktonic vs. Attached Cells

Via the second FTIR spectra analysis method, we conducted a supervised classification procedure, as described in the Materials and Methods Section. Prior to unsupervised clustering and supervised classification, we employed extra-trees in order to identify the important/significant wavenumbers of the whole spectra. It turns out that only 45 wavenumbers (essentially corresponding to two major chemical classes, as discussed later on) are adequate for efficient and robust classification and clustering purposes. Therefore, using these 45 wave numbers (shown at Figure 5 as red vertical lines), we were able to build a classification model (SVM classifier with linear kernel, see Materials and Methods Section) that reached 100% accuracy in the five-fold cross validation training phase and also at the test/validation phase (Table 1), ensuring that no over-fitting occurred. 

Continuing our analysis, and in order to support our findings and arguments stated earlier, we also conducted a literature search concerning the selected groups of wavenumbers. It is important to mention that the FTIR spectral analysis consists of a quantitative approach to study the metabolic fingerprint and can mainly detect the types of functional groups present in the structure of the molecules. More invasive and time consuming methods are used to identify the exact chemical composition, such as gas chromatography coupled to mass spectrometry. The FTIR spectra analysis, in our case, gave fast real time screening to reveal chemical group compounds and suggest—based on known recoded spectra—the corresponding molecules implicated in biochemical changes that lead to biofilm formation. From this point of view, the FTIR spectra were analyzed to further investigate the compounds, which differentiated the two groups of planktonic and attached cells on the stainless steel surface in order to better understand the mechanism behind this process (Figure 5).

The observed bands at 2960–3005 cm^−1^ correspond to free amino acids and CH stretching [44]. The numerous absorption peaks at the 2800–2950 cm^−1^ region are related to the CH_3_-CH_2_-CH strength, containing lipid compounds. More precisely, the stretching vibration of the lipid hydrocarbon tail was observed at 2921 and 2854 cm^−1^ (CH_2_), as well as at 2960 and 2869 cm^−1^ (CH_3_) [45]. A shoulder around 2938 cm^−1^ was also present that was mainly assigned to the CH_2_ stretching of ergosterol [46]. The peak at 2356 cm^−1^ was associated with carbon dioxide, while the peak at 1390 cm^−1^ was associated with the O-H bending of the phenol ring. The 900 to 1250 cm^−1^ region was typical for polysaccharides due, in part, to C-O-C and C-O ring-stretching vibrations, as well as to the P=O stretching of phosphodiesters [47,48]. More precisely, the peak at 962 cm^−1^ corresponded to the pyranose ring and the larger region at 1017 to 1045 cm^−1^, corresponds to the saccharide component ring strength. The component bands that were disclosed mainly concern β-glucans at 994 cm^−1^ (β (1–6)), at 1026 cm^−1^ (β (1–4)) and mannans at 1052 cm^−1^ [49]. 

## 4. Discussion

*B. bruxellensis* is a wine spoilage yeast species characterized by a high level of persistence in winery environments due to its biofilm formation capacity on both biotic and abiotic surfaces. The characteristic of this species to contaminate wine cellars and at the same time displaying high resistance to cleaning antimicrobial agents, is a cause of concern for the wine industry [9,50]. In our present study we firstly showed that *B. bruxellensis* strains were discriminated for their capacity to form biofilms on stainless steel surfaces, as has already been observed for other phenotypic traits with oenological interest [40,51,52,53]. The lactic acid bacteria of wine, especially the *Oenococus oeni* species, have the ability to form biofilms on stainless steel surfaces, while the attached cells contribute to the aromatic profile of the wine [54].

It is necessary for food quality and safety reasons to estimate and assess strain variability, and the related spoilage potential, in order to implement reasonable safety measures [55]. The remarkable adaptation capacity of *B. bruxellensis* is reflected via its genomic plasticity and polymorphisms, which consequently lead to expanded strain variability, while inevitably categorizing the strains vis a vis their spoilage potential [25,56]. Nevertheless, the favorable growth environment of the wine cellar promotes the invasion and surface adhesion of microorganisms [48] and at the same time shapes the genetic partners of the well adapted species [57,58]. A similar scenario has been proposed for *Saccharomyces cerevisiae*, where essential genes for adherence (FLO genes) have been maintained in wild lineages, indicating that biofilm formation is important for yeast survival in the wild [59,60,61]. 

Interestingly, the biochemical changes that lead to the cells forming biofilms were revealed for the first time by applying FTIR, a non-invasive and easy to use technique. Our innovative approach succeeded in discriminating between the spectra of planktonic vs. attached cells, as well as revealing the metabolic fingerprint implicated in biofilm formation. To date, studies have focused mainly on the proteome of attached cells of bacteria, by revealing protein regulation factors for improved adaptation to hostile conditions [62,63]. Respectively, for foodborne pathogens, it has been assumed that the biofilm formation capacity is part of the species dissemination, as the biofilm related genes have been shown to be present in all the tested strains, irrespective of other phenotypes [64]. 

Unsupervised clustering and supervised classification of the obtained spectra succeed in revealing the two main chemical groups that discriminated the attached vs. the planktonic cells: the polysaccharide and lipid groups. The presence of polysaccharides in the biofilm matrix has been well documented in the literature, especially in the case of bacteria. Polysaccharides are the primary material of the EPS matrix that shape the biofilm structure and also participate in cell attachment to surfaces [12]. In particular, according to our results, the presence of β-glucan bonding could signify the liberation procedure of the cell wall polysaccharides (β 1–6) or excreted polysaccharides (β 1–4) [65]. Indeed, the presence of β-glucan has been proven to assist the cell attachment of Botrytis cinerea on grapes [66], while this polymer can modulate cell adhesion to biotic and abiotic surfaces of wine lactic acid bacteria [67]. In the case of lipids, the adhered bacteria can decrease their membrane fluidity by altering the membrane lipid composition and better adapt to environmental conditions. Additionally, it has been documented that biofilms generally possess distinct sterol patterns in diverse phases compared with planktonic cells, and in particular, an increased amount of ergosterol was determined in the early stages of biofilm formation [68]. 

The excreted molecules, in addition to their role in cell attachment, also serve as signaling molecules to help in the communication between microbial communities. This mechanism, often referred as quorum sensing (QS), is involved in the regulation of biofilm formation, and specifically, in the conformation of the EPS structure [69]. The regulatory system of QS includes intracellular and extracellular signaling that establishes a complicated mechanism that is poorly understood [13]. It is evident that identifying the key compounds of the EPS matrix and unravelling the implicated biosynthetic pathways, will constitute an important step in managing biofilm formation processes. The present work has succeeded in representing an initial approach to study the mechanism behind the biofilm formation capacity of wine spoilage yeast, identifying more feasible a more adapted techniques to prevent future yeast adhesion and contamination during the winemaking process.

## 5. Conclusions

The results obtained from the present study reveal that the biochemical changes that occur during biofilm formation seem to be homogenous for *B. bruxellensis*, suggesting a common adaptation strategy within the species. Exploiting this outcome, we were able to develop a prediction model for efficiently and reliably assessing the phenotype of planktonic vs. attached cells, for specific strains and cultures. Furthermore, by revealing the metabolites implicated in the adhesion process, this study represents the first approach to better understanding the mechanisms behind yeast persistence in wine production cellars. Our novel developed technique will assist in the future development of preventive strategies against spoilage yeast.

## Figures and Tables

**Figure 1 microorganisms-09-00587-f001:**
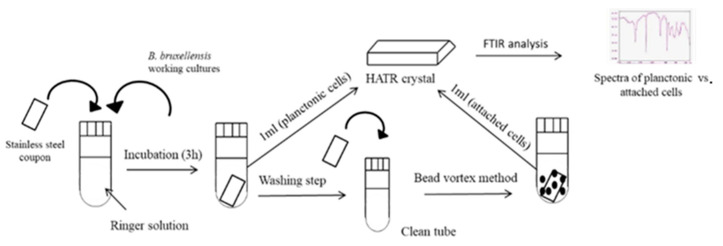
Schematic representation of the main steps implicated in samples preparation for the FTIR analysis. HATR: Horizontal Attenuated Total Reflectance

**Figure 2 microorganisms-09-00587-f002:**
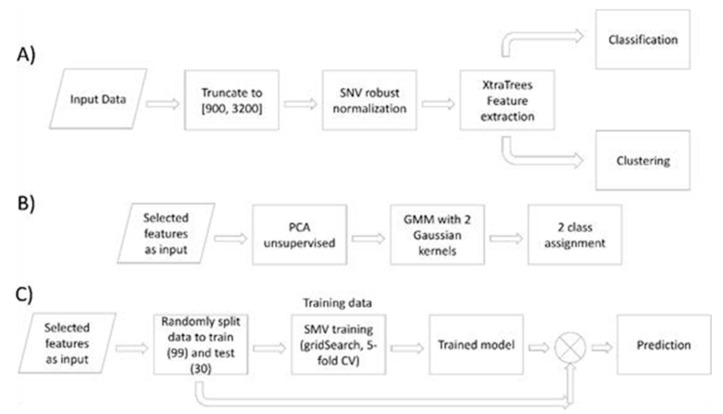
Overview of the data analysis and exploration. (**A**) Data manipulation steps towards classification and clustering. (**B**) Clustering workflow, natural clusters identification. (**C**) Classification pipeline, training and validation. SNV: standard normal variate; PCA: principal component analysis.

**Figure 3 microorganisms-09-00587-f003:**
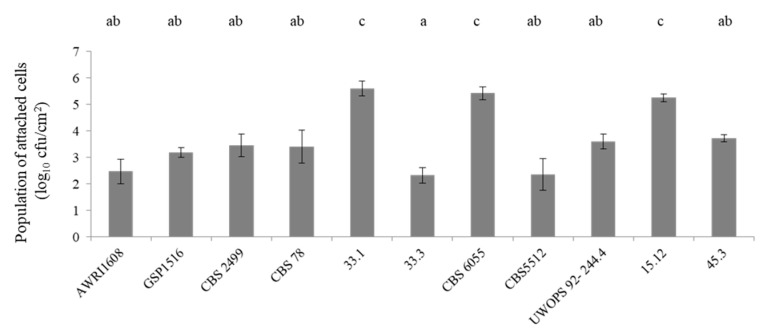
Mean population of attached cells (± standard deviation, *n* = 6) on stainless steel coupons of 11 strain s of *B. bruxellensis* in Ringer solution. Values with different superscript roman letter (a–c) in the same row are significantly different according to Tukey’s post hoc test (*p* <  0.05).

**Figure 4 microorganisms-09-00587-f004:**
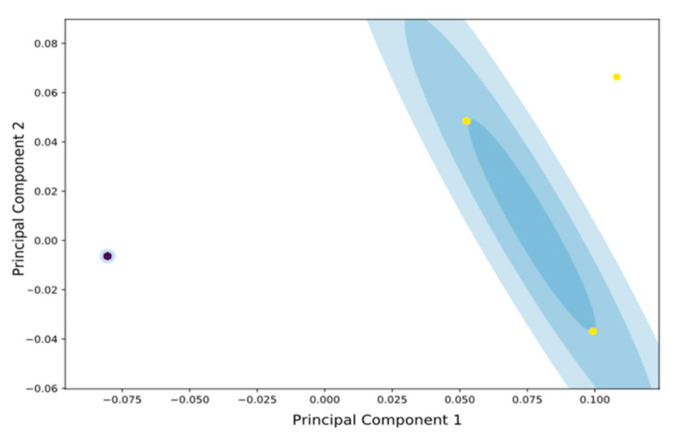
Gaussian mixture modeling of the samples of 11 *B. bruxellensis* strains in the 6 independent cultures considered. Yellow points represent the samples from planktonic cells, while blue represent attached cells. The blue shaded areas represent the Gaussian distributions corresponding to each cluster. For the planktonic cells, the distribution presents larger variance in comparison to the biofilm cell state.

**Figure 5 microorganisms-09-00587-f005:**
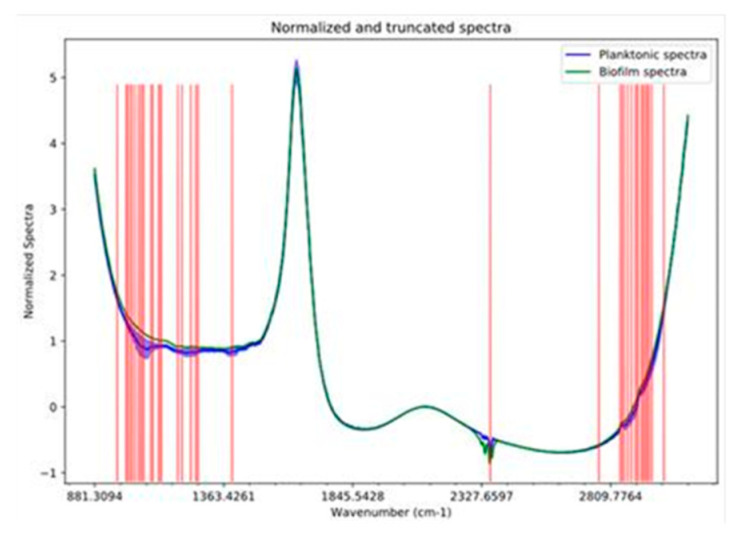
Fourier-transform infrared spectroscopy of attached and planktonic cells supernatant of 11 strains and 6 independent cultures of *B. bruxellensis*. Vertical red lines indicate the important and selected wavenumbers.

**Table 1 microorganisms-09-00587-t001:** Confusion matrix of the test samples for the trained SVM classifier.

		Actual Class (Cell State)	
		Planktonic	Attached
**Predicted class** **(cell state)**	**Planktonic**	14	0
	**Attached**	0	16

## Data Availability

The data presented in this study are available on request from the corresponding author. The data are not publicly available due to fact that explicit information and instructions should be provided.

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
