# Peer review of "Assessing the Biofilm Formation Capacity of the Wine Spoilage Yeast Brettanomyces bruxellensis through FTIR Spectroscopy"

_microorganisms, 2021, doi:10.3390/microorganisms9030587_

Round 1

Reviewer 1 Report

The manuscript titled “Assessing the biofilm formation capacity of the wine spoilage yeast Brettanomyces bruxellensis through FTIR spectroscopy" by Dimopoulou and others  succeeded for the first time to apply a non-invasive technique to reveal the metabolic fingerprint implicated in biofilm formation capacity of B. bruxellensis, underlying the homogenous mechanism within the yeast species. 

The paper merits to be published because it gives many and exhaustive information about this topic. Overall the paper is well structured and written, the conclusions are supported by the analysis of the data presented.

Author Response

Response: We would like to thank the reviewer for careful and thorough reading of this manuscript.

Reviewer 2 Report

The biofilm formation capacity of several  B. bruxellensis strains on stainless steel surfaces in aqueous solution was monitored for an incubation period of 3 hours. The statistical comparison separated the strains based on their adhesion capacity in three groups . Following  analysis via unsupervised clustering  the authors identified 2 distinct populations, forming two “natural” inherent classes that correspond to the distinction of attached cells vs. the planktonic cells. This result showed that attached cells exhibit a much less deviation among them even when the input spectra  originate from different strains and cultures.Also,planktonic cells exhibit larger variability.FTIR spectra analysis of attached vs planktonic cells of B. bruxellensis was used and finally the authors conduct a chemical compound group analysin in order to discriminate  planktonic vs attached cells. As Known it is crucial  for food quality and safety reasons to estimate and encounter strain variability and the related spoilage potential. In this study the occured biochemical changes that lead the cells to biofilm formation by applying FTIR were revealed for the first time. The authors   succeeded to discriminate the spectra of planktonic vs attached cells as well as to reveal the metabolic fingerprint implicated in biofilm formation. The researchers showed   that the biochemical changes occuring during biofilm formation seem to be homogenous for B. bruxellensis suggesting a common adaptation strategy within the species and so then ,they 
develop a prediction model for efficient and reliable for phenotype assessment, planktonic vs. attached cells, for the specific strain and culture.

This is a well wrtiten outstanding paper which contains innovative ideas for  estimating food safety.

My suggestion is to ACCEPT and publish the paper in its current form

Author Response

Response: We would like to thank the reviewer for the careful and thorough reading of this manuscript.

Reviewer 3 Report

The paper reports the biofilm formation of 11 B. bruxellensis strains to stainless steel coupons, after 3 hours of incubation in aqueous solution.  Moreover, the authors compared the metabolic fingerprint of the attached vs the planktonic cells. Finally, the chemical groups implicated in biofilm formation process were identified.

The manuscript is interesting and could give the scientific community a better and useful insight of the topic. Moreover, the authors should solve some criticisms to improve results and discussion before publication.

The introduction should be improved considering the following bibliography Tofalo R., Schirone M., Corsetti A., Suzzi G. (2012). Detection of Brettanomyces spp. in red wines using real-time PCR. Journal of Food Science, 77, M545-M549

Materials and methods section should be reduced. The authors should report references for methods used, and briefly describe the procedures.

Discussion should be improved with a better focus on the novelty of the paper. The authors should compare their results with similar studies focusing on other non-Saccharomyces species

Conclusions should be rewritten. The authors should emphasized the novelty of their work and its possible application.

Author Response

  1. The introduction should be improved considering the following bibliography Tofalo R., Schirone M., Corsetti A., Suzzi G. (2012). Detection of Brettanomyces spp. in red wines using real-time PCR. Journal of Food Science, 77, M545-M549

Response: Thank you for your suggestion. You are right, we should take under consideration that the detection of the yeast species under real oenological conditions could be complicated as the species could enter in VBNC state. Relevant bibliography has been added in the Introduction Section.

  1. Materials and methods section should be reduced. The authors should report references for methods used, and briefly describe the procedures.

Response: According to the related literature, this is the first time that such an approach is employed in terms of biofilm/plactonic classification using FT-IR spectra, not only in wine samples but also in other products/materials. We believe that the description of the methodology is as brief (or not brief) as it should since we need, apart from presenting the whole analysis pipeline as clear as possible, to also justify the use of the specific algorithms and techniques along with the “expected” outcomes from their application. This way, in this manuscript we will not only present the biological related findings but also to inform the readers on the analysis methodology, how and why it is applied. Then the readers will be able to adjust and or adapt our pipeline to other situations (in terms of type of products or other spectroscopy methods) according to their research interests.

  1. Discussion should be improved with a better focus on the novelty of the paper. The authors should compare their results with similar studies focusing on other non-Saccharomyces species

Response: In wine, besides Saccharomyces cerevisiae and Brettanomyces bruxellensis, there is no work, at least to our knowledge, concerning the biofilm formation capacity by other wine yeasts. Nevertheless, there is interesting research on the subject which focuses on the lactic acid bacteria. So as the reviewer correctly proposes to make a reference also on other species, the relative sentences have been added in the discussion part. In addition there are references on foodborne pathogens microorganisms also in the Discussion part.

  1. Conclusions should be rewritten. The authors should emphasized the novelty of their work and its possible application.

Response: Thank you for your comment, the Conclusion has been modified.

Reviewer 4 Report

L. 62-66: The aim of this work should be focused also on the identification of Brett, that is very important in the enological sector

L. 80:  The exponential phase in YPD can be reached before the 72 h, why so long here?

L. 92-93: Are these beads non-destructive for cell wall and vitality of the cells?

L. 236: I am not sure if the ringer solution can be considered a solution for incubation....please check it.

L. 254-256:  Authors think that this distinction can be useful for tracing Brett in food and particularly in wine? Please explain it.

L. 331:  Authors should know that the most important problem of Brett in enology is in wood barrels. How can Authors think to check the presence of Brett in wood, using this technique? is it possible?

L. 355-377:  All these sentences are perfect, but Authors should enphasize the role of this technique in the prevention of Brett in the biofilm form, the most resistant, should cite other foods not only wine where Bret are considered a problem.

L. 381-383: See my previous comment

Author Response

Reviewer 4

  1. 62-66: The aim of this work should be focused also on the identification of Brett, that is very important in the enological sector

Response: Thank you for your suggestion. You are right, we should take under consideration that the detection of the yeast species under real oenological conditions could be complicated as the species could enter in VBNC state. Relevant bibliography has been added in the Introduction Section.

  1. 80: The exponential phase in YPD can be reached before the 72 h, why so long here?

Response: Thank you for your comment. According to our Material and Method section (§2.2) the activation culture in YPD medium lasted for 48h while the working solution for 72h respectively. As strain variability has been observed for the growth kinetics of the species (please refer to Dimopoulou et al., 2019), 72h have been chosen in order to well synchronize the physiology state of all the 11 used strains.

Dimopoulou, M.; Hatzikamari, M.; Masneuf-Pomarede, I.; Albertin, W. Sulfur Dioxide Response of Brettanomyces Brux-ellensis Strains Isolated from Greek Wine. Food Microbiology 2019, 78, 155–163, doi:10.1016/j.fm.2018.10.013.

  1. 92-93: Are these beads non-destructive for cell wall and vitality of the cells?

Response: Thank you for your interesting question. The bead vortex method has been well tested with various microorganisms in order to study the biofilm formation capacity. Additionally no DNA release was observed in the medium after the application of the described method in the present study.

  1. 236: I am not sure if the ringer solution can be considered a solution for incubation....please check it.

Response: Thank you for your comment. Ringer has been previously used as an incubation solution for the study of B. bruxellensis biofilm formation capacity (please refer to Lebleux et al., 2020).

Lebleux, M.; Abdo, H.; Coelho, C.; Basmaciyan, L.; Albertin, W.; Maupeu, J.; Laurent, J.; Roullier-Gall, C.; Alexandre, H.; Guilloux-Benatier, M.; et al. New Advances on the Brettanomyces Bruxellensis Biofilm Mode of Life. International Journal of Food Microbiology 2020, 318, 108464, doi:10.1016/j.ijfoodmicro.2019.108464.

  1. 254-256: Authors think that this distinction can be useful for tracing Brett in food and particularly in wine? Please explain it.

Response: We think that the referred lines (254-256) of the reviewer comment are not correct but the comment is really interesting. In fact the distinction of planktonic vs the attached cells will not exactly serve to trace Brett in wine (or other food) in the terms of identification but will help to better understand the mechanism behind the adhesion capacity that provokes the invasion and contamination of the cellar. Consequently new methods of control and prevention of the attachment process could be developed. A phrase to better explain our state has been added.

  1. 331: Authors should know that the most important problem of Brett in enology is in wood barrels. How can Authors think to check the presence of Brett in wood, using this technique? is it possible?

Response: There is no doubt that the most important problem for the wine sector is the development of Brett in the pores of barrels during aging, especially when the free doses of sulphur dioxide have been decreased and when the barrels have been already used for 2, 3 or even more times. For the moment the biofilm capacity on wood surface has been tested for the species of LAB, O. oeni. So, a similar technique on wood surface can’t be excluded but possible obstacles (such as heterogeneity of the material) should be well considered.

  1. L. 355-377: All these sentences are perfect, but Authors should enphasize the role of this technique in the prevention of Brett in the biofilm form, the most resistant, should cite other foods not only wine where Bret are considered a problem.
  2. 381-383: See my previous comment

Response: The Discussion part mainly focuses on wine as the species of Brettanomyces bruxellensis is clearly considered as spoilage only in wine. For instance, in other fermented beverages such as beer Brettanomyces could act also as beneficial yeast which contributes to flavor complexity. Additional text to emphasize our findings as kindly proposed by the reviewer has been added in the Discussion and in the Conclusion Section.